# Evaluation and Comparison of Oral Health Status between Spanish and Immigrant Children Residing in Barcelona, Spain

**DOI:** 10.3390/children9091354

**Published:** 2022-09-04

**Authors:** Ana Veloso Duran, Blanca Framis-de-Mena, Maria Carmen Vázquez Salceda, Francisco Guinot Jimeno

**Affiliations:** 1Pediatric Dentistry Department, Faculty of Dentistry, Universitat Internacional de Catalunya (UIC), 08195 Barcelona, Spain; 2Department of Orthodontics, Faculty of Dentistry, Universitat de Barcelona (UB), 08007 Barcelona, Spain

**Keywords:** oral pathology, caries, dental treatments, oral health, immigrants, Spaniards

## Abstract

The present study aimed to evaluate and compare the level of oral health among Spanish and immigrant children residing in Barcelona, Spain. Oral health status was evaluated in 1400 children aged 3 to 14 years between September 2018 and June 2019. Multiple variables (dental caries lesions, exodontia, trauma, malocclusions, gingivitis, fillings, stainless steel crowns in primary dentition, and pit and fissure sealings in permanent dentition) were compared in both populations. Statistically significant differences (*p* < 0.001) were found in the prevalence of caries in the primary dentition, which was higher in the immigrant group (62.3%) than in the Spanish group (42.6%). For the permanent dentition, the prevalence of dental caries lesions was 12.2% in Spanish children and 16.4% in immigrant children, showing statistically significant differences (*p* = 0.026) between the two groups. The prevalence of fillings in the primary dentition was 14.6% in Spanish and 12.5% in immigrant children (*p* = 0.253). Regarding the permanent dentition, the number of fillings was also higher in the Spanish population (6.8%) compared to the immigrant population (3.5%), again evidencing statistically significant differences (*p* = 0.006). Our findings suggest a better oral health status in Spanish children than in immigrant children.

## 1. Introduction

Oral health is part of the general health of all individuals and should be accessible to everyone, regardless of age, ethnicity, sex, or economic status, as it is essential for a good quality of life [1,2,3,4,5]. According to the Oral Health Survey conducted in Spain in 2020, the frequency of dental visits is increasing in all groups when comparing the data obtained in previous surveys, which represents an improvement in oral health care [6].

Among the pathologies found in the oral cavity, dental caries is the most prevalent, chronic, and non-communicable disease worldwide [7,8,9]. The World Health Organization (WHO) indicates that 60–90% of children are affected by dental caries [10], with this being one of the main causes of premature tooth loss in children, causing damage to the primary dentition and future permanent dentition [7,11].

Caries is defined as a sugar-dependent disease that causes the destruction of dental tissue due to the presence of organic acids that are produced by cariogenic bacteria located in the dental biofilm, added to an imbalance in the remineralization and demineralization process over time [1,9,12,13]. In addition, other factors are involved in the development of new caries lesions and the progression of existing ones, such as the susceptibility of the host, dental hygiene, frequency of dental check-ups and behavioral, social and/or cultural factors [7,13]. It represents a public health problem and affects the quality of life of the child, and can cause problems in development, growth, and learning. It has also been associated with other medical conditions such as diabetes, obesity, and cardiovascular disease [3,12,13].

Children form part of the population that is especially susceptible to caries disease, and whether the child receives good oral care will depend largely on the parents and/or caregivers [14]. Studies [7,12] over the years have shown that the oral health of children is related to that of their parents, observing that those who have good oral health have children with better oral habits and better oral health. In addition, children who suffer caries in early childhood have serious consequences such as an increased risk of malocclusion and caries lesions in future permanent dentition [12]. Therefore, the promotion of oral health in children under five years of age is essential to maintaining health and the correct development of the functions of digestion, phonation, and ventilation [5].

In the literature, different authors have described a higher prevalence of oral pathology among immigrant populations compared to native populations and worse access to medical care for immigrants living in different countries of the world [10,15,16,17,18,19]. In Spain, many studies have recently been carried out on the prevalence of caries lesions in different child populations [6,9,20,21]. However, there are few studies [19,22,23,24] that compare oral health status between a population of immigrant children and Spanish children. Some are not recent studies, and others have been carried out by means of self-assessment surveys without performing a physical examination of the oral cavity. For this reason, the aim of the present study was to evaluate and compare oral health status by an oral examination of Spanish and immigrant children residing in Barcelona, Spain.

## 2. Materials and Methods

### 2.1. Study Design and Population

This epidemiological, descriptive, and observational study was approved by the Research Ethics Committee in Primary Health Care Jordi Gol i Gorina, Barcelona, Spain (IDIAP JGol) on 30 September 2009 (P09/85).

A minimum random sample of 1386 subjects (693 autochthonous and 693 immigrants) was determined to detect a difference between groups of 7.5% same-size by groups, accepting an alpha risk of 0.05, a beta risk of 0.2, and a 0% of loss to follow-up.

Each participant required a dental check-up, and parents/legal guardians were previously informed about the nature of the study and gave their informed consent. All the individuals agreed to participate in the study and met the following inclusion criteria: children of both sexes between 3 and 14 years of age who came for their dental check-up at the Primary Care Health Centers (CAPs) integrated into the public health service of Barcelona. The CAPs included in the study were those belonging to the SAP Litoral (Primary Care Service) in Barcelona (Casc Antic, Raval Nord, Raval Sud, Besós, La Pau, Poble Nou, Sant Martí, La Mina and Verneda) between September 2018 and June 2019. An immigrant was defined as a child born outside Spain and residing in Spain for less than 3 years, and an autochthonous child was defined as a child born in Spain or one born outside Spain but residing in Spain for at least 3 years. Children with disabilities, systemic and/or neurological diseases, the presence of syndromes, and conditions that could be associated with an increase in oral pathology were excluded from the study. The study was conducted in accordance with the Helsinki Declaration (last updated October 2013) and with the International Conference on the Guide to the Harmonization of Good Clinical Practice.

### 2.2. Procedure

The clinical examination was carried out in the dentistry service of the different CAPs using an intraoral mirror, air to dry the teeth, and light from the dental equipment. The following variables for oral health status were evaluated, and all data were recorded in a Microsoft Excel^®^ program: Dental caries lesions of deciduous/primary dentition (DD) and permanent dentition (PD) were determined as the presence or absence of caries lesions. The presence of caries was considered from white spots to deep cavities;Extraction of primary teeth and permanent teeth;Dental trauma in the primary and permanent dentition: only trauma with loss of visible tooth structure on the day of the oral examination was recorded due to lack of access to the patient’s confidential medical records;Malocclusions in the vertical plane, open bite, and deep bite were considered; in the transverse plane, we considered the presence of posterior crossbite; and in the sagittal plane, Angle’s classification was used [25];Gingivitis: the presence of local inflammation associated with a bacterial plaque in at least three teeth on some surface;Primary and permanent tooth fillings (composite, amalgam, and ionomer restorations);Stainless steel crowns on primary teeth;Pit and fissure sealings on permanent teeth.

All oral examinations were performed by five qualified and calibrated dentists. Interexaminer reliability was evaluated by analyzing 20 cases (who did not participate in the study); finding a 100% interexaminer agreement (Kappa index = 1).

### 2.3. Statistical Analysis

The statistical program R Version 4.1.1 (R Foundation for Statistical Computing. Crore Team, Vienna, Austria) was used for the tabulation and analysis of the data. The results of the evaluations were presented in the form of a distribution, according to the frequency and percentage of the variables studied. A descriptive analysis of the data was performed, analyzing the categorical variables via the Chi-square test and the numerical variables via the Student’s *t* test. *p* ≤ 0.05 was considered statistically significant.

## 3. Results

The study sample included 1400 children: 694 Spaniards and 706 immigrants residing in Spain (733 boys and 667 girls), with a mean age of 7.93 years (SD = 2.49). Table 1 describes the distribution of the sample, taking into account the sociodemographic characteristics of the children.

Regarding dental caries lesions, we observed a higher prevalence of caries in the primary dentition in immigrant children (61.9%) when compared to Spanish children (42.4%), finding statistically significant differences (*p* < 0.001), and regarding permanent dentition, more dental caries lesions were observed in immigrant children (16.4%) as compared with Spanish children (12.2%), also evidencing statistically significant differences (*p* = 0.026). Table 2 compares the oral health status of both samples, Spaniards and immigrants, in the different age groups. In the case of dental caries lesions in the permanent dentition, an increase with age was observed for both populations. Taking into account the group aged 12 to 14 years, where most of the participants had complete permanent dentition, statistically significant differences (*p* = 0.009) were observed between the two groups, with a prevalence of dental caries lesions in the permanent dentition of 69.1% among immigrants and 33.8% among Spaniards. According to the results, the age range with the highest number of dental caries lesions in the primary dentition for the immigrant children group was 6–8 years, while in Spanish children, this occurred in the 9–11 years-old sector. Extractions of primary teeth were more prevalent among immigrants (29.9%) than among Spaniards (18.4%), again showing statistically significant differences (*p* < 0.001). Taking age into account, the highest number of extractions of primary teeth for both populations was observed in the group aged 9–11 years. Regarding the number of extractions of permanent teeth, the prevalence was very similar, with Spaniards presenting 2.6% and immigrants 2.3%, without obtaining statistically significant differences (*p* = 0.691). However, at 12–14 years of age, where the dentition was largely complete permanent dentition, statistically significant differences were observed (*p* < 0.001) in terms of exodontia of permanent teeth, with a higher number among immigrants (25%) than among Spaniards (4.9%). Regarding trauma, both in the primary and permanent dentition, the immigrant group was more affected than the Spanish group; however, the differences were not statistically significant (*p* = 0.243 and *p* = 0.264). The age of the children in the two populations in the case of trauma, both in the primary and permanent dentition, did not significantly influence the results. Regarding the children who presented malocclusions, a higher percentage of immigrant children (49.7%) was observed in comparison with the Spanish children (27.8%), obtaining statistically significant differences (*p* < 0.001). According to the age of the patients for all groups, a higher percentage of children with malocclusion was observed among the immigrants compared to the Spaniards and, as age increased, the number of malocclusions increased in the immigrant population; however, among the Spaniards, it increased with age until 9–11 years of age, where the highest percentage of Spanish children with malocclusion was found. Gingivitis was also observed more in immigrants (21%) than in Spaniards (7.2%), showing statistically significant differences (<0.001). As with malocclusions, gingivitis increased at all ages among immigrants compared to Spaniards, and an increase with age was observed in both groups studied.

In reference to treatment, both restorative treatments (fillings in DD and DP and stainless-steel crowns in DD) and preventive treatments (pit and fissure sealings) predominated among native children with respect to immigrant children; however, only the difference between stainless steel crowns in primary teeth (*p* = 0.044) and between fillings in permanent teeth (*p* = 0.006) was statistically significant (Table 2).

As for the immigrant children, Table 3 shows the prevalence of dental caries lesions in both primary and permanent dentition according to their place of origin. The highest prevalence of dental caries lesions in both dentitions was observed in the group of immigrants from Asia: 40.5% in DD and 39.7% in PD.

Table 4 shows the different oral pathologies and treatments in relation to sex, with no statistically significant differences (*p* > 0.05) between boys and girls for most of the variables evaluated, except in the case of DD trauma (*p* = 0.033), where boys had a prevalence of 4.1% and girls 2.1%, and in the prevalence of PD dental caries (*p* = 0.043), where boys had a prevalence of 12.6% and girls 16.3%.

## 4. Discussion

Children’s oral health is one of the foundations on which dental care and preventive education must be built to allow for a lifetime opportunity free from preventable oral diseases [26].

The American Academy of Pediatric Dentistry (AAPD) indicates that the recommended age for the first dental visit is between 6 months of age and no later than 12 months of age [27]. However, numerous authors in different parts of the world [13,28,29,30,31,32] have revealed low rates of first dental visits at preschool age. Paglia et al. [13] state that 60% of Italian children aged 24 months had never been to the dentist. Chhabra et al. [31] observed that very few parents (15.2%) are aware that the first dental visit should be before the first birthday, and ElKarmi et al. [30] showed that only 2.6% of parents are aware of this. In Spain, the 2007 Oral Health Survey of preschoolers also revealed a low rate of visits to the dentist at an early age; specifically, at 3 years of age, 73.4% of children had not visited the dentist, and at 4 years of age, 69.5% had not visited the dentist. This early dental care should be promoted by all those health professionals who are in contact with the child patient from their first days of life and, in this way, avoid public health problems that affect the quality of life of those children [3,12,13]. However, Colombo et al. [12] state that only 1% of parents reported that they visited the dentist on the recommendation of their pediatrician or other health professional.

Over the years, the literature has shown that the oral health of the immigrant population is worse than that of the native population worldwide [17,33]. Numerous studies [4,6,19,22,23,24] have evaluated the oral health of immigrant populations residing in Spain with respect to native populations. Regardless of the methodology used in the different studies, worse oral health is observed in immigrant children compared to Spaniards [6,19,22,23,24]. Some of these studies have been carried out by means of self-assessment surveys on oral health status [4,24]. However, the present study was carried out by means of oral examinations by dentists, allowing us to make a diagnosis with certainty about the child population studied.

In accordance with the scientific evidence, a higher prevalence of dental caries lesions has also been observed in the immigrant population compared to the Spanish population [3,6,19,22,23,24]. Like these authors, the results of this study also showed a higher prevalence of dental caries in the primary dentition in immigrant children (61.9%) compared to Spanish children (42.4%) and, similarly, in the permanent dentition, obtaining 60% caries lesions in immigrants and 32.5% in Spanish children between the ages of 12 and 14 years. These values are comparable to those found by Almerich et al. [23], who obtained a prevalence of caries in Spanish children at 12 years of age, with 40.6% compared to 71.4% obtained in immigrant children, and 54.4% compared to 76.9% at 14 years of age. Paredes Gallardo et al. [19] also showed similar results with a dental caries prevalence of 47.23% in immigrants and 32.05% in Spaniards in the primary dentition, and 53.19% versus 35.34% in the permanent dentition. The high prevalence of dental caries lesions in both populations is directly related to poor eating habits, such as excessive sugar consumption and lack of or deficient oral hygiene. García Pola et al. [3] observed that 65% of immigrant children and 95.3% of Spanish children consumed sugary foods more than four times a week and, as for hygiene, it was very poor in both the immigrant group (58.8%) and the Spanish group (84.9%).

Taking age into account, Riatto et al. [22] and the results of the present study agree that the prevalence of dental caries in DD is higher at younger ages compared to the prevalence of caries in PD in older children, which may be due to the early and aggressive pattern of early childhood caries. On the other hand, in the permanent dentition, the prevalence of caries is observed to increase with increasing age. These results may be related to the fact that the consumption of sugar through carbonated beverages or ultra-processed foods is introduced as children grow older. In order to improve eating habits, Paglia et al. [13], in the period from 1 to 6 years, state that a healthy eating and drinking practice should be established based on the prevention of negative health effects, both in later childhood and in adulthood. Comparing the prevalence of dental caries lesions according to the place of origin for immigrants, the present study observes a higher dental caries prevalence among children from Asia compared to other places of origin, as does Hoyvik et al. [34]. These authors [34] state that excessive sugar consumption is often not controlled in Eastern worldwide regions, which may be one of the explanations for the higher prevalence of dental caries lesions.

Regarding the rates of gingivitis obtained in the present investigation, immigrants present a higher prevalence than Spaniards (21% vs. 7.2%), increasing as the age of the population increases. Riatto et al. [22] and Portero de la Cruz et al. [24] have evaluated the presence of gingival bleeding, observing a slightly higher percentage among immigrants than among natives.

On the other hand, the low rates of restorative (fillings and metal crowns) and preventive (pit and fissure sealings) treatments in the children in the present study, both for immigrants and Spaniards, indicate the poor knowledge of parents about the importance of correct care of the primary dentition.

Malocclusions in the present study were observed more in the immigrant children than in the Spanish children, finding statistically significant differences (*p* < 0.001) between both populations. Despite this statistically significant difference, both groups presented a high prevalence; therefore, it is important to inform parents that the prolonged use of pacifiers and/or bottle feeding, as well as the habit of digital sucking, have a negative influence on the correct development of the oral cavity, and can lead to malocclusions, such as anterior open bite or posterior cross bite, among other complications [35,36].

Regarding the influence of the child’s sex on oral health status, the main research studies [19,22,23,24] concluded that sex does not significantly influence oral health status, as did the results obtained for most of the variables evaluated in the present research study, with the exception of trauma in DD (*p* = 0.033) and caries lesions in PD (*p* = 0.043). In any case, the results of our study in relation to the frequency of dental trauma cannot be conclusive since, at the time of collecting the information, we did not have access to the patient’s clinical record, so it was impossible to collect those traumas that had occurred in the past, as well as those in which there was no loss of dental structure, which could produce a bias in the information.

There are some limitations to the present research study. The socioeconomic and cultural status of the parents of the participants was not recorded, although oral health can be affected by these variables; therefore, it would be interesting to have these data available for future research. Also, the dental examination could be accompanied by a questionnaire referring to data such as when the first visit to the dentist was made, the frequency of periodic check-ups, and the daily consumption of sugars.

## 5. Conclusions

The oral health status of immigrant children living in Barcelona, Spain, is worse than that of Spanish children in global terms;The number of dental caries lesions in the primary dentition prevailed among immigrant children in most age groups, and the number of dental caries lesions in the permanent dentition prevailed among immigrant children of all ages;The prevalence of both restorative and preventive treatments performed was higher in Spanish children than in immigrant children for most age groups except in the case of stainless crowns at 3–5 years old and sealings at 6–8 years old, where the prevalence was higher among immigrants;In the range of 3–5 years old, statistically significant differences were observed between both populations, with a higher prevalence among immigrants in the case of caries lesions in the primary dentition, malocclusions, and gingivitis;In the 12–14-year range, statistically significant differences were observed between both populations, with a higher prevalence among immigrants in the case of permanent dentition caries lesions and malocclusions;Sex, regardless of whether the child is an immigrant or native, does not influence most of the variables evaluated for oral health status;Due to the caries rate in both populations continuing to be high, it is necessary to implement early care programs for the infant population and to make parents aware of the importance of oral cavity care and good oral health from birth.

## Figures and Tables

**Table 1 children-09-01354-t001:** Sociodemographic characteristics of the population in the present research study.

Sociodemographic Characteristics			*n* (%)
Age	Spaniards	3–5 years	61 (8.8%)
		6–8 years	355 (51.2%)
		9–11 years	197 (28.4%)
		12–14 years	81 (11.7%)
	Inmigrants	3–5 years	112 (15.9%)
		6–8 years	378 (53.5%)
		9–11 years	172 (24.4%)
		12–14 years	44 (6.2%)
Sex	3–5 years	Boy	89 (12.1%)
		Girl	84 (12.6%)
	6–8 years	Boy	381 (52%)
		Girl	352 (52.8%)
	9–11 years	Boy	200 (27.3%)
		Girl	169 (25.3%)
	12–14 years	Boy	63 (8.6%)
		Girl	62 (9.3%)
Origin of immigrants		Asia	246 (34.8%)
		South America	225 (31.9%)
		Africa	109 (15.4%)
		Central America	49 (6.9%)
		Oceania	3 (0.4%)
		North America	2 (0.3%)
		Europe	72 (10.2%)

**Table 2 children-09-01354-t002:** Comparison of oral health between Spanish and immigrant children in different age ranges from 3 to 14 years.

Origin	Age Range												Global		
	3–5 Spaniards *n* = 61	3–5 Immigrants *n* = 112	*p* Value	6–8 Spaniards *n* = 355	6–8 Immigrants *n* = 378	*p* Value	9–11 Spaniards *n* = 197	9 -11 Immigrants *n* = 172	*p* Value	12–14 Spaniards *n* = 81	12–14 Immigrants *n* = 44	*p* Value	3–14 Spaniards *n* = 694	3–14 Immigrants *n* = 706	*p* Value
Prevalence of dental caries in DD	26 (43.3%)	72 (64.3%)	0.008 *	166 (46.8%)	269 (71.2%)	<0.001 *	95 (48.2%)	95 (55.2%)	0.179	7 (9.1%)	1 (2.5%)	0.180	294 (42.4%)	437 (61.9%)	<0.001 *
Prevalence of extractions in DD	9 (15%)	22 (20%)	0.420	71 (20%)	126 (33.3%)	<0.001 *	43 (21.8%)	61 (35.5%)	0.004 *	5 (6.5%)	2 (5%)	0.747	128 (18.4%)	211 (29.9%)	<0.001 *
Prevalence of trauma in DD	10 (16.7%)	19 (17.3%)	0.920	5 (1.4%)	6 (1.6%)	0.842	3 (1.5%)	1 (0.6%)	0.384	0 (0%)	0 (0%)	1	18 (2.6%)	26 (3.7%)	0.243
Prevalence of dental caries in PD	0 (0%)	0 (0%)	1	26 (7.3%)	41 (10.8%)	0.098	32 (16.2%)	49 (28.5%)	0.005 *	27 (33.8%)	26 (59.1%)	0.009 *	85 (12.2%)	116 (16.4%)	0.026 *
Prevalence of trauma in PD	0 (0%)	0 (0%)	1	1 (0.3%)	3 (0.8%)	0.347	3 (1.5%)	5 (2.9%)	0.362	3 (3.7%)	4 (9.1%)	0.211	7 (1.0%)	12 (1.7%)	0.264
Prevalence of malocclusions	8 (13.1%)	33 (29.5%)	0.016 *	100 (28.2%)	187 (49.5%)	<0.001 *	64 (32.5%)	101 (58.7%)	<0.001 *	21 (25.9%)	30 (68.2%)	<0.001 *	193 (27.8%)	351 (49.7%)	<0.001 *
Prevalence of gingivitis	1 (1.6%)	13 (11.5%)	0.022 *	13 (3.7%)	73 (19.3%)	<0.001 *	22 (11.2%)	48 (27.9%)	<0.001 *	14 (17.3%)	14 (31.8%)	0.063	50 (7.2%)	148 (21.0%)	<0.001 *
Prevalence of fillings in DD	2 (3.3%)	3 (2.7%)	0.822	58 (16.3%)	62 (16.4%)	0.981	35 (17.8%)	23 (13.4%)	0.247	6 (7.4%)	0 (0%)	0.070	101 (14.6%)	88 (12.5%)	0.253
Prevalence of stainless steel crowns in DD	0 (0%)	1 (0.9%)	0.459	7 (2%)	3 (0.8%)	0.169	7 (3.6%)	2 (1.2%)	0.138	1 (1.3%)	0 (0%)	0.469	15 (2.2%)	6 (0.8%)	0.044 *
Prevalence of fillings in PD	0 (0%)	0 (0%)	1	5 (1.4%)	4 (1.1%)	0.667	23 (11.7%)	13 (7.6%)	0.184	19 (23.5%)	8 (18.2%)	0.494	47 (6.8%)	25 (3.5%)	0.006 *
Prevalence of sealings in PD	0 (0%)	0 (0%)	1	4 (1.1%)	7 (1.9%)	0.420	9 (4.6%)	4 (2.3%)	0.244	3 (3.9%)	1 (2.5%)	0.693	16 (2.3%)	12 (1.7%)	0.418
Prevalence of extractions in PD	0 (0%)	0 (0%)	1	5 (1.4%)	3 (0.8%)	0.423	9 (4.6%)	2 (1.2%)	0.055	4 (4.9%)	11 (25.0%)	<0.001 *	18 (2.6%)	16 (2.3%)	0.691

DD = Deciduous/Primary dentition; PD = Permanent dentition; * Statistically significant value.

**Table 3 children-09-01354-t003:** Prevalence of dental caries lesions in DD and PD according to the place of origin of the immigrants.

Origin of Inmigrants	Prevalence of Dental Caries Lesions in DD *n* (%)	Prevalence of Dental Caries Lesions in PD *n* (%)
Asia	177 (40.5%)	46 (39.7%)
South America	119 (27.2%)	35 (30.2%)
Africa	82 (18.8%)	21 (18.1%)
Central America	22 (5.0%)	3 (2.6%)
Oceania	0 (0.0%)	0 (0%)
North America	0 (0.0%)	0 (0%)
Europe	37 (8.5%)	11 (9.5%)

**Table 4 children-09-01354-t004:** Prevalence of caries lesions in DD and PD according to the sex of the children.

	Age Range												Global		
	3–5 (*n* = 173)			6–8 (*n* = 733)			9–11 (*n* = 369)			12–14 (*n* = 125)			3–14 (*n* = 1400)		
	Boy *n* = 89	Girl *n* = 84	*p* Value	Boy *n* = 381	Girl *n* = 352	*p* Value	Boy *n* = 200	Girl *n* = 169	*p* Value	Boy *n* = 63	Girl *n* = 62	*p* Value	Boy *n* = 733	Girl *n* = 667	*p* Value
Prevalence of dental caries in DD	52 (59.8%)	45 (54.2%)	0.465	240 (63%)	195 (55.4%)	0.036	92 (46.0%)	98 (58.0%)	0.022	4 (6.6%)	4 (7.1%)	0.900	388 (53.1%)	342 (51.3%)	0.502
Prevalence of extractions in DD	18 (20.7%)	13 (15.7%)	0.396	107 (21.8%)	90 (25.6%)	0.443	53 (26.5%)	51 (30.2%)	0.434	2 (3.3%)	5 (8.9%)	0.198	180 (24.6%)	159 (23.8%)	0.754
Prevalence of trauma in DD	17 (19.5%)	12 (14.5%)	0.379	9 (2.4%)	2 (0.6%)	0.046 *	4 (2.0%)	0 (0%)	0.065	0 (0%)	0 (0%)	1	30 (4.1%)	14 (2.1%)	0.033 *
Prevalence of dental caries in PD	0 (0%)	0 (0%)	1	30 (7.9%)	37 (10.5%)	0.216	37 (18.5%)	44 (26.0%)	0.081	25 (39.7%)	28 (45.2%)	0.535	92 (12.6%)	109 (16.3%)	0.043 *
Prevalence of trauma in PD	0 (0%)	0 (0%)	1	3 (0.8%)	1 (0.3%)	0.355	6 (3.0%)	2 (1.2%)	0.233	5 (7.9%)	2 (3.2%)	0.252	14 (1.9%)	5 (0.7%)	0.061
Prevalence of malocclusions	17 (19.1%)	24 (28.6%)	0.106	149 (39.1%)	138 (39.2%)	0.979	93 (46.5%)	72 (42.6%)	0.453	21 (33.3%)	30 (48.4%)	0.087	280 (38.2%)	264 (39.6%)	0.596
Prevalence of gingivitis	9 (10.1%)	5 (6.0%)	0.316	51 (13.4%)	35 (9.9%)	0.148	41 (20.5%)	29 (17.2%)	0.415	14 (22.2%)	14 (22.6%)	0.962	115 (15.7%)	83 (12.4%)	0.082
Prevalence of fillings in DD	3 (3.4%)	2 (2.4%)	0.689	59 (15.5%)	61 (17.3%)	0.500	37 (18.5%)	21 (12.4%)	0.110	6 (9.8%)	0 (0%)	0.016 *	105 (14.3%)	84 (12.6%)	0.344
Prevalence of stainless steel crowns	1 (1.1%)	0 (0%)	0.330	4 (1.0%)	6 (1.7%)	0.445	6 (3.0%)	3 (1.8%)	0.447	1 (1.6%)	0 (0%)	0.336	12 (1.6%)	9 (1.3%)	0.658
Prevalence of fillings in PD	0 (0%)	0 (0%)	1	8 (2.1%)	21 (6.0%)	0.007 *	20 (10.0%)	18 (10.7%)	0.838	19 (30.2%)	11 (17.7%)	0.104	47 (6.4%)	50 (7.5%)	0.425
Prevalence of seals in PD	0 (0%)	0 (0%)	1	5 (1.3%)	6 (1.7%)	0.663	8 (4.0%)	5 (3.0%)	0.589	3 (4.9%)	1 (1.8%)	0.352	16 (2.2%)	12 (1.8%)	0.609
Prevalence of extractions in PD	0 (0%)	0 (0%)	1	3 (0.8%)	5 (1.4%)	0.410	11 (5.5%)	0 (0%)	0.002 *	7 (11.1%)	8 (12.9%)	0.758	21 (2.9%)	13 (1.9%)	0.266

* Statistically significant value.

## Data Availability

The data presented in this study are available on request from the corresponding author. The data are not publicly available due to ethical requirements.

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
