# Peer review of "Evaluation and Comparison of Oral Health Status between Spanish and Immigrant Children Residing in Barcelona, Spain"

_children, 2022, doi:10.3390/children9091354_

Round 1

Reviewer 1 Report

Review

A Study on the Evaluation and Comparison of Oral Health Status between 2 Spanish and Immigrant Children Residing in Barcelona, Spain presents a topical issue. The evaluation of the odontal, periodontal status, of the oro-dental health condition being a necessity for the forecasting of the health programs. In order to publish this article, some changes are needed.

ABSTRACT

1.     Abstract: The material and method section, results, discussion and conclusion are presented systematically. I suggest that age should be homogenized. 1389 subjects are passed here, in results 1400.

INTRODUCTION

2.     The introduction can be improved, the health assessment being carried out in all the countries of the European Union and in the whole world (see WHO database). It is interesting to note the prevalence of dental caries in other countries by age groups.

MATERIAL AND METHOD

3.     The study was approved in 2009 by the Research Ethics Committee in Primary Health Care Jordi Gol i Gorina, Barcelona, ​​Spain (ID-IAP JGol) on September 30, 2009 (P09 / 85) and was conducted in 2018-2019, and other previous components, or was the opinion simply requested 10 years ago before application?

4.     The authors did not present how the sampling was performed and whether each sample by its age group is representative.

5.     In the Procedures section we discussed the evaluation of dental caries in the ICDAS system (Line 95-97) but in the results there is no individualization by codes. This is very important. If you have performed the evaluation in this system, the results must also appear on the codes. Code 1 or 2 may require primary preventive care, the other codes require other therapeutic approaches. The same goes for seals. Cannot be mixed with composite restorations, amalgam…

6.     How were the malocclusions evaluated, with what kind of index?

7.     The study group was larger in the 3-6 age group (almost double) for immigrants and in the 12-14 age group half that of the Spanish children. This aspect can greatly influence the results. The authors pointed out that they proposed a gender equality in the 650/650 estimate, but why did they not propose an equality between the number of subjects by age categories between the studied groups?

8.     What index was used to assess periodontal health to determine the level of gingivitis. Is it localized, generalized, has it been evaluated quantitatively or qualitatively?

9.     How was the sampling done?

10.  I believe that 3–6-year-olds should be separated from 6–12-year-olds in the study.

DISCUSSIONS

11.  In this section lines 248-254 and 265-267 should be removed because the study does not refer to oral hygiene habits.

12.  In this study the level of knowledge of the parents was not evaluated, therefore the paragraph line 265-267 should be deleted. Because children have been evaluated for routine check-ups at the Primary Preventive Care Clinic, where they receive primary care, are immigrant children not given the same facilities, namely tooth sealing?

13.  Were there any differences between the children in choosing the option to seal their teeth? Especially since the caries risk of immigrant children was higher.

14.  With regard to paragraphs 268-274, have children 's vicious habits been assessed in order to correlate them with malocclusion? Please explain this.

CONCLUSIONS

15.  Reformulate the conclusions by age groups, the 3–6-year-old group being double in favor of immigrant children and the 12–14-year-old group being double in favor of Spanish children.

16.  For permanent dentition, children after the age of 6 can be considered.

17.  Draw conclusions on caries codes.

BIBLIOGRAPHY

The bibliography is up to date, no self-quotations are highlighted. The bibliography respects the writing norms of the scientific journal

After rewriting the analyzed aspects, I will review.

Author Response

REVIEWER 1:

First of all, we would like to thank you for your contribution to the manuscript and the time you spent on it. We have tried to resolve all the issues raised.

A Study on the Evaluation and Comparison of Oral Health Status between 2 Spanish and Immigrant Children Residing in Barcelona, Spain presents a topical issue. The evaluation of the odontal, periodontal status, of the oro-dental health condition being a necessity for the forecasting of the health programs. In order to publish this article, some changes are needed.

ABSTRACT

  1. Abstract: The material and method section, results, discussion and conclusion are presented systematically. I suggest that age should be homogenized. 1389 subjects are passed here, in results 1400.

Sorry, this was our fault. The sample determined was of a minimum random sample of 1386 subjects calculated by the statistics service of the International University of Catalonia before starting the study, however, the final sample comprised 1400 subjects. We have change it in the manuscript at line 13.

INTRODUCTION

  1. The introduction can be improved, the health assessment being carried out in all the countries of the European Union and in the whole world (see WHO database). It is interesting to note the prevalence of dental caries in other countries by age groups.

We have added this bibliography here to respond to your comment about the importance

of knowing the prevalence of dental caries in different countries. We have not added it

in the text because we believe that despite being something important, there is a lot of

literature on the high prevalence of caries and we believe that in our introduction may

be excessive information. However, if you think it is important to specify it in the

manuscript, we will add it in the introduction without any problem.

- Valpreda, L.;Carcieri, P.; Cabras, M.; Vecchiati, G.; Arduino, PG.; Bassi, F.

Frequency and severity of dental caries in foster care children of Turin, Italy: a

retrospective cohort study. Eur. J. Paediatr. Dent. 2020, 21, 299-302.

- Yassin, S.M.; Tikare, S.; AlKahtani, Z.M.; AlFaifi, F.J.; AlFaifi, W.S.: AlFaifi, E.;

Omair, A.; Ravi, K.S. Caries preventive practices and dental caries among boys aged 6

15 in Saudi Arabia. Eur. J. Paediatr. Dent. 2020, 21, 97-102.

- Obregón-Rodríguez, N.; Fernández-Riveiro, P.; Piñeiro-Lamas, M.; Smyth-Chamosa,

E.; Montes-Martínez, A.; Suárez-Cunqueiro, M.M. Prevalence and caries-related risk

factors in schoolchildren of 12- and 15-year-old: a cross-sectional study. BMC. Oral.

Health. 2019, 19,120.

- Pérez, B.M.; Silla, A.J.; Santos, G.G.; et al. Encuesta de Salud Oral en España 2020. RCOE: Revista del Ilustre Consejo General de Colegios de Odontólogos y Estomatólogos de España. 2020, 25, 14-20

- Almerich-Torres, T.; Montiel-Company, J.M.; Bellot-Arcís, C.; Iranzo-Cortés, J.E.;

Ortolá-Siscar, J.C.; Almerich-Silla, J.M. Caries Prevalence Evolution and Risk Factors

among Schoolchildren and Adolescents from Valencia (Spain): Trends 1998-2018. Int.

  1. Environ. Res. Public. Health. 2020, 17, 6561.

- Bissar, A.R.; Schulte, A.G.; Muhjazi, G.; Koch, M.J. Caries prevalence in 11- to 14

year old migrant children in Germany. Int. J. Public. Health. 2007, 52, 103-108.

MATERIAL AND METHOD

  1. The study was approved in 2009 by the Research Ethics Committee in Primary Health Care Jordi Gol i Gorina, Barcelona, ​​Spain (ID-IAP JGol) on September 30, 2009 (P09 / 85) and was conducted in 2018-2019, and other previous components, or was the opinion simply requested 10 years ago before application?

The study was approved in 2009 by the Jordi Gol i Gorina Primary Health Care Research Ethics Committee, Barcelona, ​​Spain (ID-IAP JGol) on September 30, 2009 (P09/85) as part of a protocol with different lines of research, which have been carried out during this time in different phases. Some of the phases were published: Veloso A, López Giménez J, Vázquez MC, Corcuera JR, Guinot F, Puigdollers A. Relationship between the order of permanent tooth eruption and the predominance of motor function laterality: a cross-sectional study. An Pediatr (Engl Ed). 2021 Jun;94(6):396-402. doi: 10.1016/j.anpede.2020.12.004. Epub 2021 Jan 22. PMID: 34090636. This study was carried out in 2018-2019 when the methodology of the present study was conducted.

  1. The authors did not present how the sampling was performed and whether each sample by its age group is representative.

The CAPs included in the study were those belonging to the SAP Litoral (Primary Care Service) of Barcelona (Casc Antic, Raval Nord, Raval Sud, Besós, La Pau, Poble Nou, Sant Martí, La Mina and Verneda), which represents a representative part of the city of Barcelona.

Having totally homogeneous groups could be better but due to the complexity of the accessibility of the different age groups to primary care facilities and after consultation with the statistical service it was established that the groups could be compared between them.

I leave here the contact e-mail and the name of the manager in case you have any questions:- Juan Carlos Martín- [email protected] ORCID: 000-0002-1045-4802

  1. In the Procedures section we discussed the evaluation of dental caries in the ICDAS system (Line 95-97) but in the results there is no individualization by codes. This is very important. If you have performed the evaluation in this system, the results must also appear on the codes. Code 1 or 2 may require primary preventive care, the other codes require other therapeutic approaches. The same goes for seals. Cannot be mixed with composite restorations, amalgam…

The ICDAS system was used only to let the examiners know what to consider caries and what not. All values ​​from 1 to 6 were considered as presence of caries and value 0 as absence of caries. Not every caries lesion with ICDAS code was recorded. We found other studies which consider too, the prevalence of caries as presence and absence.

- Obregón-Rodríguez, N.; Fernández-Riveiro, P.; Piñeiro-Lamas, M.; Smyth-Chamosa,

E.; Montes-Martínez, A.; Suárez-Cunqueiro, M.M. Prevalence and caries-related risk

factors in schoolchildren of 12- and 15-year-old: a cross-sectional study. BMC. Oral.

Health. 2019, 19,120.

- da Silva SN, Gimenez T, Souza RC, Mello-Moura ACV, Raggio DP, Morimoto S,

Lara JS, Soares GC, Tedesco TK. Oral health status of children and young adults with

autism spectrum disorders: systematic review and meta-analysis. Int J Paediatr Dent.

2017 Sep;27(5):388-398. doi: 10.1111/ipd.12274. Epub 2016 Oct 31. PMID: 27796062.

We will take it into account for future researchs.

As for the restorations, all those of composite, amalgam and ionomer were considered as such, apart from the sealings of pits and fissures.

We have clarified and modified it in the article in lines 109-110.

  1. How were the malocclusions evaluated, with what kind of index?

Any alteration concerning the patient's occlusion diagnosed in the three planes of space: transverse, vertical and/or sagittal.

In the vertical plane, openbite and overbite were considered as malocclusions; in the transverse plane it was the presence of posterior crossbite; and in the sagittal plane Angle’s classification was used. Angle E. Classification of Malocclusion. Dental Cosmos. 1899. 74 (248-264); 350-357.

We have clarified and modified it in the article in lines 105-107.

  1. The study group was larger in the 3-6 age group (almost double) for immigrants and in the 12-14 age group half that of the Spanish children. This aspect can greatly influence the results. The authors pointed out that they proposed a gender equality in the 650/650 estimate, but why did they not propose an equality between the number of subjects by age categories between the studied groups?

The statistical service established that the groups could be compared between them after determine the sample.

However it is true that having totally homogeneous groups in terms of age groups would have been better at interpreting the results. We think too that the fact of dividing the sample by age groups is to offer the reader more precise and enriching information.

I leave here the contact e-mail and the name of the manager in case you have any questions:- Juan Carlos Martín- [email protected] ORCID: 000-0002-1045-4802

  1. What index was used to assess periodontal health to determine the level of gingivitis. Is it localized, generalized, has it been evaluated quantitatively or qualitatively?

We have determined as gingivitis the presence of local inflammation associated with presence of bacterial plaque withouth an index. The lack of means and time in the Spanish public primary Health care services didn’t allow to carry out the plaque control. Other authors evaluated the gingivitis as any deviation from normality of the tissues.

  1. da Silva SN, Gimenez T, Souza RC, Mello-Moura ACV, Raggio DP, Morimoto S, Lara JS, Soares GC, Tedesco TK. Oral health status of children and young adults with autism spectrum disorders: systematic review and meta-analysis. Int J Paediatr Dent. 2017 Sep;27(5):388-398. doi: 10.1111/ipd.12274. Epub 2016 Oct 31. PMID: 27796062.
  2. How was the sampling done?

We have evaluated all the children of both sexes between 3 and 14 years old who attended their dental check-up at the Primary care Health centers (CAP). The children with disabilities, systemic and/or neurological diseases, presence of syndromes and conditions that could be associated with an increase in oral pathology were excluded. The CAPs included where those belonging to SAP Litoral of Barcelona (CAPs that belonged to the same area of the city), being a representative sample of the city.

  1. I believe that 3–6-year-olds should be separated from 6–12-year-olds in the study.

We have considered that is interesting to consider the preschool age since early childhood caries has an aggressive pattern and higher prevalence whole world and that’s why we think it can enrich the study a lot.

Other studies reviewed and compared at the discussion they also cover preschool ages, we leave here some examples:

- Riatto, S.G.; Montero, J.; Pérez, D.R.; Castaño-Séiquer, A.; Dib, A. Oral Health Status

of Syrian Children in the Refugee Center of Melilla, Spain. Int. J. Dent. 2018, 18,

2637508, doi: 10.1155/2018/2637508. (Includes from 5 years to 13 years)

- Kale, S.; Kakodkar, P.; Shetiya, S.; Abdulkader, R. Prevalence of dental caries among

children aged 5-15 years from 9 countries in the Eastern Mediterranean Region: a meta

analysis. East. Mediterr. Health. J. 2020, 26, 726-735. (Includes from 5 years to 15

years)

- Valpreda, L.; Carcieri, P.; Cabras, M.; Vecchiati, G.; Arduino, PG.; Bassi, F.

Frequency and severity of dental caries in foster care children of Turin, Italy:

retrospective cohort study. Eur. J. Paediatr. Dent. 2020, 21, 299-302. (Includes from 4

years to 12 years)

- Portero de la Cruz, S.; Cebrino, J. Oral Health Problems and Utilization of Dental

Services Among Spanish and Immigrant Children and Adolescents. Int. J. Environ. Res.

Public. Health. 2020, 17, 738. (Includes from 3 years to 15 years).

DISCUSSIONS

  1. In this section lines 248-254 and 265-267 should be removed because the study does not refer to oral hygiene habits.

We believe that it is important to take into account and talk about the frequency of brushing in children since gingivitis is a disease induced by plaque and health professionals must be made aware of the importance of brushing properly from childhood since we observe in the literature low frequency levels.Finally, if the reviewer believes it is advisable to delete this phrase, we will remove it from the manuscript

  1. In this study the level of knowledge of the parents was not evaluated, therefore the paragraph line 265-267 should be deleted. Because children have been evaluated for routine check-ups at the Primary Preventive Care Clinic, where they receive primary care, are immigrant children not given the same facilities, namely tooth sealing?.

We think it is important to know the low knowledge of the parents about toothbrushing to enrich the study and raise awareness among Health professionals who are in contact with Pediatric patients and thus achieve good prevention programs.

Both immigrants and autochthonous children received the same facilities at the primary Health care centers, however not all the centers can offer the same services since some do not have the means to offer preventive treatments such as seals.

Finally, if the reviewer believes it is advisable to delete this phrase from the text, we will eliminate it without any problem.

  1. Were there any differences between the children in choosing the option to seal their teeth? Especially since the caries risk of immigrant children was higher.

Each Primary Care Health Centers (CAP) due to the limitation of budget and infrastructure can offer some treatments or others. Seals are offered as a preventive treatment, in the same way to immigrants and autochthonous in those teeth without caries to avoid the appearance of the same, however as we said previously not all the primary health care centers offer the same treatments.

  1. With regard to paragraphs 268-274, have children 's vicious habits been assessed in order to correlate them with malocclusion? Please explain this.

The vicious habits have not been evaluated in this population due to the lack of information especially in the immigrant population. We eliminate this information since in the study sample was very incomplete due to the language problems or because they didn’t remember, especially in the older ages. With these paragraphs at the discussion, we are attempting to give a scientific explanation based on published studies.

CONCLUSIONS

  1. Reformulate the conclusions by age groups, the 3–6-year-old group being double in favor of immigrant children and the 12–14-year-old group being double in favor of Spanish children.

We have changed at the text in lines 298-305.

“The oral health status of immigrant children living in Barcelona, Spain, is worse than those of Spanish children in global terms.

The number of dental caries lesions in primary dentition prevailed among immigrant children in most age groups and the number of dental caries lesions in permanent dentition prevailed among immigrant children in all ages.

The prevalence of treatments performed, both restorative and preventive, was higher in Spanish children than in immigrant children in most age groups except in the case of stainless crowns at 3-5 years and sealings at 6-8 years where the prevalence was higher among immigrants.”

  1. For permanent dentition, children after the age of 6 can be considered.

We have evaluated the caries lesions according to the dentition since a percentatge of primary dentition caries lesions can be detected in children over 6 years when the patient has mixed dentition.

  1. Draw conclusions on caries codes.

We only recorded the presence or absence of caries without evaluating the severity of the caries lesions. Thats why we don’t have the caries codes.

BIBLIOGRAPHY

The bibliography is up to date, no self-quotations are highlighted. The bibliography respects the writing norms of the scientific journal

After rewriting the analyzed aspects, I will review.

Reviewer 2 Report

Dear authors, thank you for your work. I have several concerns.

To diagnose caries, the International Caries Detection and Diagnosis System (ICDAS) was used: All lesions with an ICDAS index of 1 to 6 are considered caries lesions. This system implies a detailed reflection of the intensity (depth) of the carious process. However, in the results, caries is reflected only as the presence/absence. Why did the authors not describe the index data and its coding in any way, this would allow one to track the intensity of caries in the two studied groups in more detail?

The authors evaluated the presence/absence of gingivitis in children, but did not assess the level of bacterial plaque (its amount and location), while it is the bacterial factor that is one of the leading factors in the development of caries and periodontal disease. Why was it decided not to evaluate this parameter?

In the Introduction, it is postulated that "different authors have described in the literature a higher prevalence of oral pathology among immigrant populations compared to native populations and a worse access to medical care by immigrants in different countries of the world [10,15-19]." What then is the scientific novelty and value of this particular study? Except for the fact that it was completed in a more recent time frame?

Maybe it was more correct to 1) divide the children into groups according to the types of dentition: primary, mixed, permanent, and 2) to use classical indices to determine the intensity of caries (dmft, PMA, GI index of ortho need and other) and 3) to compare the obtained data with other studies, which would allow for an estimated comparison of the data of this study with others?

Therefore, I can conclude that the manuscript in its present form lacks the scientific novelty and the results cannot be generalized.

Author Response

REVIEWER 2:

First of all, we would like to thank you for your contribution to the manuscript and the time you spent on it. We have tried to resolve all the issues raised.

Dear authors, thank you for your work. I have several concerns.

  1. To diagnose caries, the International Caries Detection and Diagnosis System (ICDAS) was used: All lesions with an ICDAS index of 1 to 6 are considered caries lesions. This system implies a detailed reflection of the intensity (depth) of the carious process. However, in the results, caries is reflected only as the presence/absence. Why did the authors not describe the index data and its coding in any way, this would allow one to track the intensity of caries in the two studied groups in more detail?

The ICDAS system was used only to let the examiners know what to consider caries and what not. All values ​​from 1 to 6 were considered as presence of caries and value 0 as absence of caries. Not every caries lesion with ICDAS code was recorded. We found other studies which consider too, the prevalence of caries as presence and absence.

  1. Obregón-Rodríguez, N.; Fernández-Riveiro, P.; Piñeiro-Lamas, M.; Smyth-Chamosa, E.; Montes-Martínez, A.; Suárez-Cunqueiro, M.M. Prevalence and caries-related risk factors in schoolchildren of 12- and 15-year-old: a cross-sectional study. BMC. Oral. Health. 2019, 19,120.
  2. da Silva SN, Gimenez T, Souza RC, Mello-Moura ACV, Raggio DP, Morimoto S, Lara JS, Soares GC, Tedesco TK. Oral health status of children and young adults with autism spectrum disorders: systematic review and meta-analysis. Int J Paediatr Dent. 2017 Sep;27(5):388-398. doi: 10.1111/ipd.12274. Epub 2016 Oct 31. PMID: 27796062.

We will take it into account for future researchs.

Regarding the fillings, all composite, amalgam and ionomer restorations were considered other tan sealants.

  1. The authors evaluated the presence/absence of gingivitis in children, but did not assess the level of bacterial plaque (its amount and location), while it is the bacterial factor that is one of the leading factors in the development of caries and periodontal disease. Why was it decided not to evaluate this parameter?

We have determined as gingivitis the presence of local inflammation associated with presence of bacterial plaque withouth an index. The lack of means and time in the Spanish public primary Health care services didn’t allow to carry out the plaque control. Other authors evaluated the gingivitis as any deviation from normality of the tissues.

  1. da Silva SN, Gimenez T, Souza RC, Mello-Moura ACV, Raggio DP, Morimoto S, Lara JS, Soares GC, Tedesco TK. Oral health status of children and young adults with autism spectrum disorders: systematic review and meta-analysis. Int J Paediatr Dent. 2017 Sep;27(5):388-398. doi: 10.1111/ipd.12274. Epub 2016 Oct 31. PMID: 27796062.

  1. In the Introduction, it is postulated that "different authors have described in the literature a higher prevalence of oral pathology among immigrant populations compared to native populations and a worse access to medical care by immigrants in different countries of the world [10,15-19]." What then is the scientific novelty and value of this particular study? Except for the fact that it was completed in a more recent time frame?

 There are studies around the world that have described a poor oral health among immigrant populations compared to native ones. However, in Spain there are few studies and the majority have been carried out by self-surveys about oral health without an oral examination. Our study has been carried out with oral examination in a dental box with mirror, air to dry de teeth and light of the dental equipment. We think it is interesting too since the higher prevalence of caries still existing and the low rates of restoration in child population which implies that parents have low knowledge about the importance of their children’s oral health. We believe that this study can provide a lot of important information in order to improve the state of oral health.

  1. Maybe it was more correct to 1) divide the children into groups according to the types of dentition: primary, mixed, permanent, and 2) to use classical indices to determine the intensity of caries (dmft, PMA, GI index of ortho need and other) and 3) to compare the obtained data with other studies, which would allow for an estimated comparison of the data of this study with others?

We thought it was more interesting to divide the sample into groups according the age as we have reviewed other previous studies in the literature that divided the sample according to the age of the children and we think that this enriches the information of the study and make it more precise. Also, we divided according to primary dentition and permanent dentition all the evaluated variables. We didn’t determine the index of the caries lesions because we only noted if there was presence or absence of caries and the way to consider presence of caries was the ICDAS index (1-6 values were considered as presence of caries and 0 was considered as absence of caries).

We leave here some articles that have divided the sample by age groups.

  1. Riatto, S.G.; Montero, J.; Pérez, D.R.; Castaño-Séiquer, A.; Dib, A. Oral Health Status of Syrian Children in the Refugee Center of Melilla, Spain. Int. J. Dent. 2018, 18, 2637508, doi: 10.1155/2018/2637508.
  2. Almerich-Torres, T.; Montiel-Company, J.M.; Bellot-Arcís, C.; Iranzo-Cortés, J.E.; Ortolá-Siscar, J.C.; Almerich-Silla, J.M. Caries Prevalence Evolution and Risk Factors among Schoolchildren and Adolescents from Valencia (Spain): Trends 1998-2018. Int. J. Environ. Res. Public. Health. 2020, 17, 6561.
  3. Bissar, A.R.; Schulte, A.G.; Muhjazi, G.; Koch, M.J. Caries prevalence in 11- to 14-year old migrant children in Germany. Int. J. Public. Health. 2007, 52, 103-108.
  4. Gibbs L, de Silva AM, Christian B, Gold L, Gussy M, Moore L, Calache H, Young D, Riggs E, Tadic M, Watt R, Gondal I, Waters E. Child oral health in migrant families: A cross-sectional study of caries in 1-4 year old children from migrant backgrounds residing in Melbourne, Australia. Community Dent Health. 2016 Jun;33(2):100-6.

  1. Therefore, I can conclude that the manuscript in its present form lacks the scientific novelty and the results cannot be generalized.

Most of the studies carried out in Spain that compare the oral health status of immigrants and Spaniards have been carried out through a survey about the oral health status without carrying out an oral examination. Also, there are not many recent studies.

Despite the access to information, the prevalence of caries remains high, as well as the low rate of treatment, that’s why we believe it is important to carry out the study to help raise awareness about the importance of having good oral health from childhood.

Reviewer 3 Report

First of all, thank you for the opportunity to review this manuscript.

The purpose of this study is to was to evaluate and compare the level of oral health among Spanish and immigrant children residing in Barcelona, Spain.

The idea of study is very interesting, but realization has same shortcomings. The following are suggestions for the present manuscript:

Methodology

1. The criteria methods used in the study should be the method recommended by the WHO. Please clarify how to diagnose dental caries and the instruments used here.

2. Please add the sample size calculation. The authors wrote that they used 650 participants per group? Why? How many immigrants’ children aged 3-14 are in Barcelona? The sample size needs to be calculated based on immigration data? Or calculate the Cohen's d based on the results between groups.

3. Also, it is imperative to know how many years the child has lived in Spain. It is important to compare now immigrants and those who live longer in Spain. Are the new immigrants more vulnerable to caries than those who live longer in the new country?

4. Also, what is about dmft index and SiC Index. Please calculate it. And see the difference between newcomers and locals.

Results

1. Table 1 is unclear. What are all the demographics taken? Why wasn't the family income or parents' education level? The table compares these data between newcomers and the local population.

Author Response

REVIEWER 3:

First of all, we would like to thank you for your contribution to the manuscript and the time you spent on it. We have tried to resolve all the issues raised.

First of all, thank you for the opportunity to review this manuscript.

The purpose of this study was to evaluate and compare the level of oral health among Spanish and immigrant children residing in Barcelona, Spain.

The idea of study is very interesting, but realization has same shortcomings. The following are suggestions for the present manuscript:

Methodology

1. The criteria methods used in the study should be the method recommended by the WHO. Please clarify how to diagnose dental caries and the instruments used here.  The clinical caries was diagnose using the International Caries Detection and Diagnosis System (ICDAS) and all the lesions with an ICDAS index from 1 to 6 were considered as presence of caries and ICDAS 0 was considered as absence of caries, not the severity of them. The oral examination was carried out with an intraoral mirror, air to dry the teeth and light from the dental equipment. The WHO states that a complete examination of all the teeth requires a mirror and a good illumination of the oral cavity.

  1. Please add the sample size calculation.

A minimum random sample of 1386 subjects (693 autochthonous and 693 immigrants) was determined to detect a difference between groups of 7,5%, same size by groups, 0% of loss to follow-up. Finally the total sample comprised 1400 subjects. We have changed at the text.    

  1. The authors wrote that they used 650 participants per group? Why?

The sample was determined by the statistics service of the International University of Catalonia prior to starting the study, determining that a minimum random sample of 1,386 subjects (693 natives and 693 immigrants) was needed to detect a differencebetween groups of 7,5%, same size by groups, 0% of loss to follow-up. However, the final sample consisted of 1,400 subjects (694 Spaniards and 706 immigrants residing in Spain (733 boys and 667 girls), with a mean age of 7.93 ( SD = 2.49). All the statistics have been carried out with the support of the Statistics Service of the Universitat Internacional de Catalunya. I leave here the contact e-mail and the name of the manager in case you have any questions:- Juan Carlos Martín- [email protected] ORCID: 000-0002-1045-4802

  1. How many immigrants’ children aged 3-14 are in Barcelona:

The exact number of migrant children assigned to the outpatient clinics studied is not easy to calculate since we only have data on the general population (adults and children, foreigners and nationals) assigned to the outpatient clinics involved in the study, which were 159,292, out of a general population of Barcelona of 2,239,915 inhabitants in other words, the patients in our centers represented 7.11% of the inhabitants of the region to which the city of Barcelona belongs.

These data can be compared at (https://www.idescat.cat/pub/?id=aec&n=1041&lang=es&t=2021).

However, in each of these public health centers not only the percentage of foreign population varied, but also within it the percentage of children (3 to 14 years old) in each of these centers fluctuated, in which, furthermore, already in those dates, their origin was not considered when opening the paediatric files.

However, what could be demonstrated was that the sample of the two groups was homogeneous, both in terms of age and data collection. Homogeneity that could also be verified in similar studies on equivalent populations.

  1. The sample size needs to be calculated based on immigration data? Or calculate the Cohen's d based on the results between groups:

We have calculated in detecting statistically significant differences.

  1. Also, it is imperative to know how many years the child has lived in Spain. It is important to compare now immigrants and those who live longer in Spain. Are the new immigrants more vulnerable to caries than those who live longer in the new country?

In the study, all children born in Spain and those born in any other country outside Spain who had been residing in our country for at least 3 years were considered natives and those born outside Spain that have been living in Spain for less than 3 years were considered immigrants to eliminate the bias of differences in access to public health. It is true that, at first, the study was planned based on the children's first visits to the dentistry service, but this design introduced a large number of poorly measurable variables that could influence the results, so, finally, it was chose this type of criteria and coincided with other studies.

- García-Pola M, González-Díaz A, García-Martín JM. Effect of a Preventive Oral Health Program Starting during Pregnancy: A Case-Control Study Comparing Immigrant and Native Women and Their Children. Int J Environ Res Public Health. 2021 Apr 13;18(8):4096. doi: 10.3390/ijerph18084096. PMID: 33924511; PMCID: PMC8069462.

  1. Also, what is about dmft index and SiC Index. Please calculate it. And see the difference between newcomers and locals.

There are some studies that calculate the dmft index to evaluate the prevalence of caries in different populations, however there are other ones as our that consider the prevalence of caries too only noting the prevalence with percentages of presence and absence of caries. We have considered that calculate the prevalence of caries as this way is easier to compare with other studies and is easier to communicate to the readers since not everyone is going to know the caries index. Also, we think is interesting to reach all the health professionals who have contact with the pediatric patients.

We leave here some articles that have evaluated the prevalence of caries calculating percentages of presence and absence of the lesions.

  1. Obregón-Rodríguez, N.; Fernández-Riveiro, P.; Piñeiro-Lamas, M.; Smyth-Chamosa, E.; Montes-Martínez, A.; Suárez-Cunqueiro, M.M. Prevalence and caries-related risk factors in schoolchildren of 12- and 15-year-old: a cross-sectional study. BMC. Oral. Health. 2019, 19,120.
  2. da Silva SN, Gimenez T, Souza RC, Mello-Moura ACV, Raggio DP, Morimoto S, Lara JS, Soares GC, Tedesco TK. Oral health status of children and young adults with autism spectrum disorders: systematic review and meta-analysis. Int J Paediatr Dent. 2017 Sep;27(5):388-398. doi: 10.1111/ipd.12274. Epub 2016 Oct 31. PMID: 27796062.

 Results

  1. Table 1 is unclear. What are all the demographics taken?

The age and sex of all the subjects were registered and the country of origin of the immigrant population too.

  1. Why wasn't the family income or parents' education level? The table compares these data between newcomers and the local population.

Thank you for the contribution. This would have been interesting to know. The family income or parent’s education level would be a way to complete and interpret the results of the study. However, this was difficult because since they were public centers, access to some records was complicated.

This is a good point and will be noted for future research.

Round 2

Reviewer 1 Report

Review. 2

The study saw an improvement, assessing health status for any population, being welcome because it provides multiple information that can guide dental health decision makers to take action. Unfortunately, the study does not have the evaluation tools that are validated and are currently required, respectively:

MATERIAL AND METHOD 1. The authors did not present how the sampling was performed and whether each sample by its age group is representative. Contact the statistics department for guidance on this aspect of sampling. 2. For the ICDAS index (Line 95-97) I think you should remove it because you did not evaluate by codes. Specify that you used the DMFT index (because it turns out that you used it). There are indeed studies that monitored tooth decay as present or absent, but they do not specify that they used the ICDAS index. 3. There is an IOTN index for assessing the need for orthodontic treatment depending on the severity of the malocclusion. Why didn't you use it? Please specify. 4. Regarding the paragraph “Why was the study group higher in the age group 3-6 years (almost double) for immigrants and in the age group 12-14 years half that of Spanish children? You did not answer the question. Please answer and specify in conclusions this aspect for more accuracy. 5. You specified in the comments that the Statistical Service has established that the groups can be compared with each other after determining the sample. Once again I emphasize that you did not specify how you calculated the sample. 6. The evaluation of gingival inflammation can be highlighted by clinical appearance and by palpation with periodontal probe. Gingivitis has as a determining factor the bacterial plaque but there are other favorable factors that contribute to the appearance of gingivitis. Specify in the text that the assessment of gingival status was performed by inspection and was considered as the presence of gingivitis, the presence of inflammation at least 1 tooth, or 3 teeth, or otherwise… 7. To the question "How was the sampling done?" answer with inclusion and exclusion criteria. Review. 8. Regarding the remark "I think that children aged 3-6 should be separated from those aged 6-12 in the study" you did not answer clearly. Of course, the prevalence of dental caries at the age of 3-6 years is interesting (we have mostly temporary teeth) but we do not mix it and compare it with the one at the age of 6-12 years (in dynamics we have fewer and fewer temporary teeth and more permanent fines). The study is not enriched in this way but confusing, in the sense that it does not provide relevant information and there are many biases. DISCUSSIONS 9. Regarding the remark “In this section, lines 248-254 and 265-267 should be deleted because the study does not refer to oral hygiene habits” you replied that oral hygiene is important. Of course it is important, but you did not highlight it in any way in the study. Discussing the general aspects of brushing your baby does not influence or say anything about your study. I maintain the deletion of this paragraph. 10. It is also very good to assess the level of knowledge of parents, but the authors did not highlight this aspect in any way in their study. 11. As to whether immigrant children are not offered the same facilities, namely dental sealing, you replied: 'Both immigrants and local children have received the same facilities in primary care centers, but not all centers can provide the same services, because some do not have the means to offer preventive treatments, such as sealing ”. In this sense, explain in the study in the material and method section how many children and why they were evaluated in the centers without the possibility of sealing. Basically, there was a bias regarding the choice of children for whom seals were made. If you specify no, then the choice to receive or not to seal was not influenced by the immigrant status but by the decision of the specialists. conclusions 12. Maintain -reformulate the conclusions by age groups, the group of 3-6 years being double in favor of immigrant children, and the group of 12-14 years being double in favor of Spanish children.

Author Response

First of all, thank you very much for your recommendations. We have tried to resolve all your comments and have improved the English in the manuscript.

The study saw an improvement, assessing health status for any population, being welcome because it provides multiple information that can guide dental health decision makers to take action. Unfortunately, the study does not have the evaluation tools that are validated and are currently required, respectively:

  1. The authors did not present how the sampling was performed and whether each sample by its age group is representative. Contact the statistics department for guidance on this aspect of sampling.

Within a Primary Care Centre, a pilot test of this same study was carried out with a total of 60 individuals (30 from each group) and we obtained the next prevalence of caries: 8% for the spaniards and 12,54% for the immigrants.

Based on these results, we have calculated a minimum sample size necessary for our study with the GRANMO online calculator. Accepting an alpha risk of 0,05 and a beta risk of 0,2 in a two-sided test, 693 subjects are necessary in the first group and 693 subjects in the second to find as statistically significant a proportion difference, expected to be of 0,08 in group 1 and 0,13 in group 2. It has been anticipated a drop-out rate of 0%.

Despite this sample size calculation, we have obtained 694 spaniards and 706 immigrants.

We have added in the sample size, on material and methods section, that we assume a 0,05 Alpha risk and a 0,2 Beta risk.

I leave here the contact e-mail and the name of the statistics department who has collaborated with us:

- Juan Carlos Martín, Assistant Professor of Universitat Internacional de Catalunya / [email protected] / ORCID: 000-0002-1045-4802

- Caterina Paparsenos Fernández, statistical / [email protected]

  1. For the ICDAS index (Line 95-97) I think you should remove it because you did not evaluate by codes. Specify that you used the DMFT index (because it turns out that you used it). There are indeed studies that monitored tooth decay as present or absent, but they do not specify that they used the ICDAS index.

According to their recommendations, we have removed the ICDAS index from the text since we have not evaluated the caries lesions of the population with its codes. We have described in materials and methods the way in which caries lesions have been evaluated: presence of caries or absence of caries. Caries lesion was considered from the presence of white spot to deep cavities. We didn’t evaluate the number of caries or the severity.

We have based ourselves on other studies similar to ours that evaluate the presence or absence of caries too regardless of whether they use any index:

- Obregón-Rodríguez, N.; Fernández-Riveiro, P.; Piñeiro-Lamas, M.; Smyth-Chamosa, E.; Montes-Martínez, A.; Suárez-Cunqueiro, M.M. Prevalence and caries-related risk factors in schoolchildren of 12- and 15-year-old: a cross-sectional study. BMC. Oral. Health. 2019, 19,120.

- Riatto, S.G.; Montero, J.; Pérez, D.R.; Castaño-Séiquer, A.; Dib, A. Oral Health Status of SyrianChildren in the Refugee Center of Melilla, Spain. Int. J. Dent. 2018, 18, 2637508, doi: 10.1155/2018/2637508.

  1. There is an IOTN index for assessingt the need for orthodontic treatment depending on the severity of the malocclusion. Why didn't you use it? Please specify.

We didn’t use this index because we only evaluated if there was any alteration concerning the patient occlusion diagnosed in the three planes of space: transverse, vertical and/or saggital. We used the Angle’s classification for the saggital plane because we use this classification in all the programs of our university since it is a simple way for the child patient. In the case of the vertical plane, we only considered if the patient had overbite or openbite and finally in the transverse plane, we only observed whether or not there was a cross bite. We only evaluated the presence or absence of malocclusion not the severity that’s way we didn’t use and index for assessing the need for orthodontic treatment depending on the severity of the malocclusion.

Angle E. Classification of Malocclusion. DentalCosmos. 1899. 74 (248-264); 350-357.

  1. Regarding the paragraph “Why was the study group higher in the age group 3-6 years (almost double) for immigrants and in the age group 12-14 years half that of Spanish children? You did not answer the question. Please answer and specify in conclusions this aspect for more accuracy.

The main objective of the study was to detect differences in oral health status in children aged from 3 to 14 years between the two populations in question (Spaniards and immigrants). Among the different comparisons made, we have considered the possibility of seeing if there are differences in the ages of the children in both groups. Through the Mann-Witney test, we have seen that there were significant differences (p<0,001) in the ages of patients obtaining medians of 8 and 7 years for spaniards and immigrants, respectively. Given these results, we found interesting to regroup the children in 4 age groups (3-5, 6-8, 9-11 and 12-14), to be able to make more accurate comparisons. Comparisons for each age group have been made separately for each age group. As presence/absence percentages are being compared, it is not mandatory that the number of children be identical in the two groups, considering that it has not been our main objective. It is true that having totally homogeneous groups in terms of age groups would have been better at interpreting the results. However, since our analysis is purely descriptive, we consider that dividing the sample by age groups can offer us greater accuracy and information about our data.

I leave here the contact e-mail and the name of the statistics department who has collaborated with us:

- Juan Carlos Martín, Assistant Professor of Universitat Internacional de Catalunya / [email protected] / ORCID: 000-0002-1045-4802

- Caterina Paparsenos Fernández, statistical / [email protected]

  1. You specified in the comments that the Statistical Service has established that the groups can be compared with each other after determining the sample. Once again I emphasize that you did not specify how you calculated the sample.

Within a Primary Care Centre, a pilot test of this same study was carried out with a total of 60 individuals (30 from each group) and we obtained the next prevalence of caries: 8% for the spaniards and 12,54% for the immigrants.

Based on these results, we have calculated a minimum sample size necessary for our study with the GRANMO online calculator. Accepting an alpha risk of 0,05 and a beta risk of 0,2 in a two-sided test, 693 subjects are necessary in the first group and 693 subjects in the second to find as statistically significant a proportion difference, expected to be of 0,08 in group 1 and 0,13 in group 2. It has been anticipated a drop-out rate of 0%.

Despite this sample size calculation, we have obtained 694 spaniards and 706 immigrants.

The two original populations used for the comparison are homogeneous (694 Spanish children and 706 immigrants). Patients have been randomly selected as they reached CAP centers to a total of 1,400 children. From the beginning we have made no distinction in the age, as it was not our main goal. Comparisons by age bracket have been made for a purely descriptive purpose, not for inference. These results can serve as a reference to propose a new line of study, focusing on the age groups that have shown the greatest differences according to our study.

I leave here the contact e-mail and the name of the statistics department who has collaborated with us:

- Juan Carlos Martín, Assistant Professor of Universitat Internacional de Catalunya / [email protected] / ORCID: 000-0002-1045-4802

- Caterina Paparsenos Fernández, statistical / [email protected]

  1. The evaluation of gingival inflammation can be highlighted by clinical appearance and by palpation with periodontal probe. Gingivitis has as a determining factor the bacterial plaque but there are other favorable factors that contribute to the appearance of gingivitis. Specify in the text that the assessment of gingival status was performed by inspection and was considered as the presence of gingivitis, the presence of inflammation at least 1 tooth, or 3 teeth, or otherwise…

We have evaluated the gingivitis by the inspection of the gums and the bacterial plaque. We determined as gingivitis the presence of inflammation associated with bacterial plaque in at least 3 teeth. The ideal would have been to make a plaque control in each patient as other authors to accurately and objectively assess the percentage of surfaces with bacterial plaque, but due to the lack of economic means (not all these centers have plaque control) and lack of time it was not possible.

We have specified in the manuscript how the gingivitis was determined.

  1. To the question "How was the sampling done?" answer with inclusion and exclusion criteria. Review.

We have obtained 694 spanish children and 706 immigrant children between 3-14 years old visited in the different Primary Health Centers. Each participant required a dental check-up and parents/legal guardians were previously informed about the nature of the study and gave their informed consent. All the individuals agreed to participate in the study. Children have obtained randomly provided that the inclusion and exclusion criteria were respected:

- Inclusion criteria:

- Children from both sexes.

- Children between 3 and 14 years old.

- Immigrant child: child born outside Spain and residing in Spain for less than 3 years.

- Autochthonous child: child born in Spain or born outside Spain but residing in Spain for at least 3 years.

- Exclusion criteria:

            - Children with disabilities.

            - Children with systemic and/or neurological diseases.

- Children with syndromes and conditions that could be associated with an increase in oral pathology.

Once the database has been made and purified, ensuring that all dadta has been collected the statistical analysis has been carried out.

  1. Regarding the remark "I think that children aged 3-6 should be separated from those aged 6-12 in the study" you did not answer clearly. Of course, the prevalence of dental caries at the age of 3-6 years is interesting (we have mostly temporary teeth) but we do not mix it and compare it with the one at the age of 6-12 years (in dynamics we have fewer and fewer temporary teeth and more permanent fines). The study is not enriched in this way but confusing, in the sense that it does not provide relevant information and there are many biases.

Initially, the main objective was to evaluate and compare the global one, that is, from 3 to 14 years specifying when it was temporary teething and when it was permanent teething. Then it was divided by age groups as in another study that I show below in order to give more information and to be able to evaluate the changes as the child grows.

- Riatto, S.G.; Montero, J.; Pérez, D.R.; Castaño-Séiquer, A.; Dib, A. Oral Health Status of SyrianChildren in the Refugee Center of Melilla, Spain. Int. J. Dent. 2018, 18, 2637508, doi: 10.1155/2018/2637508.

  1. Regarding the remark “In this section, lines 248-254 and 265-267 should be deleted because the study does not refer to oral hygiene habits” you replied that oral hygiene is important. Of course it is important, but you did not highlight it in any way in the study. Discussing the general aspects of brushing your baby does not influence or say anything about your study. I maintain the deletion of this paragraph.

Following his recommendations, we have deleted the paragraphs he names in order to improve the manuscript.

  1. It is also very good to assess the level of knowledge of parents, but the authors did not highlight this aspect in any way in their study.

It would be very important and very useful information for our study. However, sometimes the companions are not the parents, in the case of immigrants sometimes the language is a barrier and often this information is not routinely accessible to the professional. We will take this into account for future research as it is one of the limitations of our study.

  1. As to whether immigrant children are not offered the same facilities, namely dental sealing, you replied: 'Both immigrants and local children have received the same facilities in primary care centers, but not all centers can provide the same services, because some do not have the means to offer preventive treatments, such as sealing ”. In this sense, explain in the study in the material and method section how many children and why they were evaluated in the centers without the possibility of sealing. Basically, there was a bias regarding the choice of children for whom seals were made. If you specify no, then the choice to receive or not to seal was not influenced by the immigrant status but by the decision of the specialists. conclusions

What we wanted to say by the other answer is that all children, both autochthonous and immigrant, receive the same attention within the same centre. All the Primary Care Centers where these children have been explored offer the same type of treatment because they belong to the same area. There may be other centers in other areas where the same treatments are not provided because not everyone can offer such services. It is difficult to assess this as there are children who may have changed residence and therefore changed primary care, other children are sometimes visited in private centres and sometimes in public centres and may therefore have been sealed by pits and fissures in a private centre or others who may have appointment to make the seals and annul it and do not return. We have evaluated the presence or absence of seals, not where and why they had been sealed.

  1. Maintain -reformulate the conclusions by age groups, the group of 3-6 years being double in favor of immigrant children, and the group of 12-14 years being double in favor of Spanish children.

Although the main objective is the global one, due to his request we add the conclusions of these two age groups here and in the manuscript.

- The oral health status of immigrant children living in Barcelona, Spain, is worse than those of Spanish children in global terms.

- The number of dental caries lesions in primary dentition prevailed among immigrant children in most age groups and the number of dental caries lesions in permanent dentition prevailed among immigrant children in all ages.

- In the range of 3 to 5 years, statistically significant differences were observed between both populations with a higher prevalence among immigrants in the case of caries lesions in primary dentition, malocclusions and gingivitis.

- In the of 12 to 14 years, statistically significant differences were observed between both populations with a higher prevalence among immigrants in the case of permanent dentition caries lesions and malocclusions.

- The prevalence of treatments performed, both restorative and preventive, was higher in Spanish children than in immigrant children in most age groups except in the case of stainless crowns at 3-5 years and sealings at 6-8 years where the prevalence was higher among immigrants.

- Sex, regardless of whether the child is immigrant or native, does not influence most of the variables evaluated on oral health status.

- Due to the caries rate in both populations continues to be high, it is necessary to implement early care programs for the infant population and to make parents aware of the importance of oral cavity care and good oral health from birth.

Reviewer 2 Report

Unfortunately, no constructive responses were received.

Author Response

REVIEWER 2:

First of all, thank you very much for your recommendations and the time you spent on it. We have tried to resolve all your comments and have improved the English in the manuscript.

Dear authors, thank you for your work. I have several concerns.

  1. To diagnose caries, the International Caries Detection and Diagnosis System (ICDAS) was used: All lesions with an ICDAS index of 1 to 6 are considered caries lesions. This system implies a detailed reflection of the intensity (depth) of the carious process. However, in the results, caries is reflected only as the presence/absence. Why did the authors not describe the index data and its coding in any way, this would allow one to track the intensity of caries in the two studied groups in more detail?

We have removed the ICDAS index from the text since we have not evaluated the caries lesions of the population with its codes. We have described in materials and methods the way in which caries lesions have been evaluated: presence of caries or absence of caries. Caries lesion was considered from the presence of white spot to deep cavities. We didn’t evaluate the number of caries or the severity.We have based ourselves on other studies similar to ours that evaluate the presence or absence of caries too regardless of whether they use any index:- Obregón-Rodríguez, N.; Fernández-Riveiro, P.; Piñeiro-Lamas, M.; Smyth-Chamosa, E.; Montes-Martínez, A.; Suárez-Cunqueiro, M.M. Prevalence and caries-related risk factors in schoolchildren of 12-and 15-year-old: a cross-sectional study. BMC. Oral. Health. 2019, 19,120.- Riatto, S.G.; Montero, J.; Pérez, D.R.; Castaño-Séiquer, A.; Dib, A. Oral Health Status of SyrianChildren in the Refugee Center of Melilla, Spain. Int. J. Dent. 2018, 18, 2637508, doi:10.1155/2018/2637508. 

  1. The authors evaluated the presence/absence of gingivitis in children, but did not assess the level of bacterial plaque (its amount and location), while it is the bacterial factor that is one of the leading factors in the development of caries and periodontal disease. Why was it decided not to evaluate this parameter?

We have evaluated the gingivitis by the inspection of the gums and the bacterial plaque. We determined as gingivitis the presence of inflammation associated with bacterial plaque in at least 3 teeth. The ideal would have been to make a plaque control in each patient as other authors to accurately and objectively assess the percentage of surfaces with bacterial plaque, but due to the lack of economic means (not all these centers have plaque control) and lack of time it was not possible.

We have specified in the manuscript how the gingivitis was determined.

  1. In the Introduction, it is postulated that "different authors have described in the literature a higher prevalence of oral pathology among immigrant populations compared to native populations and a worse access to medical care by immigrants in different countries of the world [10,15-19]." What then is the scientific novelty and value of this particular study? Except for the fact that it was completed in a more recent time frame?

 There are studies around the world that have described a poor oral health among immigrant populations compared to native ones. However, in Spain there are few studies and the majority have been carried out by self-surveys about oral health without an oral examination. Our study has been carried out with oral examination in a dental box with mirror, air to dry de teeth and light of the dental equipment. We think it is interesting too since the higher prevalence of caries still existing and the low rates of restoration in child population which implies that parents have low knowledge about the importance of their children’s oral health. We believe that this study can provide a lot of important information in order to improve the state of oral health.

  1. Maybe it was more correct to 1) divide the children into groups according to the types of dentition: primary, mixed, permanent, and 2) to use classical indices to determine the intensity of caries (dmft, PMA, GI index of ortho need and other) and 3) to compare the obtained data with other studies, which would allow for an estimated comparison of the data of this study with others?

Thank you for your recommendations. We did it like this because our main objective was to evaluate and to the comparisons with the global analysis (from 3 to 14 years). However we thought it was more interesting to divide the sample into groups according the age since we have seen it in previous studies. Also, we divided according to primary dentition and permanent dentition all the evaluated variables. We didn’t determine the index of the caries lesions because we only noted if there was presence or absence of caries without evaluating the number of caries lesions or the severity as in other variables such gingivitis and malocclusions. We considered as presence all the lesions from white spots to deep cavities in the case of caries lesions. Gingivitis was determined as the presence of local inflammation associated with bacterial plaque in at least 3 teeth and finally the malocclusions were evaluated in the three planes; we used the Angle classification in the sagital plane as is a classification used in our university for the all programs, we leave below too.

We have compared the obtained data with other similar studies made in Spain or in other countries that evaluate similar variables and in the same way.

We leave here some articles that have divided the sample by age groups.

  1. Riatto, S.G.; Montero, J.; Pérez, D.R.; Castaño-Séiquer, A.; Dib, A. Oral Health Status of Syrian Children in the Refugee Center of Melilla, Spain. Int. J. Dent. 2018, 18, 2637508, doi: 10.1155/2018/2637508.
  2. Almerich-Torres, T.; Montiel-Company, J.M.; Bellot-Arcís, C.; Iranzo-Cortés, J.E.; Ortolá-Siscar, J.C.; Almerich-Silla, J.M. Caries Prevalence Evolution and Risk Factors among Schoolchildren and Adolescents from Valencia (Spain): Trends 1998-2018. Int. J. Environ. Res. Public. Health. 2020, 17, 6561.
  3. Bissar, A.R.; Schulte, A.G.; Muhjazi, G.; Koch, M.J. Caries prevalence in 11- to 14-year old migrant children in Germany. Int. J. Public. Health. 2007, 52, 103-108.
  4. Gibbs L, de Silva AM, Christian B, Gold L, Gussy M, Moore L, Calache H, Young D, Riggs E, Tadic M, Watt R, Gondal I, Waters E. Child oral health in migrant families: A cross-sectional study of caries in 1-4 year old children from migrant backgrounds residing in Melbourne, Australia. Community Dent Health. 2016 Jun;33(2):100-6.

Angle E. Classification of Malocclusion. DentalCosmos. 1899. 74 (248-264); 350-357.

  1. Therefore, I can conclude that the manuscript in its present form lacks the scientific novelty and the results cannot be generalized.

Most of the studies carried out in Spain that compare the oral health status of immigrants and Spaniards have been carried out through a survey about the oral health status without carrying out an oral examination by a professional. Also, there are not many recent studies comparing these two populations.

Despite the access to information thanks to social media and increase contact between different specialities dealing with the child patient too, the prevalence of caries remains high, as well as the low rate of treatment, that’s why we believe it is important to carry out the study to help raise awareness about the importance of having good oral health from childhood. We think these type of studies are important too in order to know whether oral health status remains the same, improves or worsens over time and thus know the usefulness of preventive measures that exist today.
